# Epilepsy Characteristics in Neurodevelopmental Disorders: Research from Patient Cohorts and Animal Models Focusing on Autism Spectrum Disorder

**DOI:** 10.3390/ijms231810807

**Published:** 2022-09-16

**Authors:** Sukanya Chakraborty, Rrejusha Parayil, Shefali Mishra, Upendra Nongthomba, James P. Clement

**Affiliations:** 1Neuroscience Unit, Jawaharlal Nehru Centre for Advanced Scientific Research, Bengaluru 560064, India; 2Molecular Reproduction, Development and Genetics (MRDG), Indian Institute of Science, Bengaluru 560012, India

**Keywords:** autism spectrum disorder, *Syngap1*, *Fmr1*, *Shank*, *Mecp2*, *Tsc1/2*, *Scna1*, biomarkers, epilepsy, electroencephalography

## Abstract

Epilepsy, a heterogeneous group of brain-related diseases, has continued to significantly burden society and families. Epilepsy comorbid with neurodevelopmental disorders (NDDs) is believed to occur due to multifaceted pathophysiological mechanisms involving disruptions in the excitation and inhibition (E/I) balance impeding widespread functional neuronal circuitry. Although the field has received much attention from the scientific community recently, the research has not yet translated into actionable therapeutics to completely cure epilepsy, particularly those comorbid with NDDs. In this review, we sought to elucidate the basic causes underlying epilepsy as well as those contributing to the association of epilepsy with NDDs. Comprehensive emphasis is put on some key neurodevelopmental genes implicated in epilepsy, such as *MeCP2*, *SYNGAP1*, *FMR1*, *SHANK1-3* and *TSC1*, along with a few others, and the main electrophysiological and behavioral deficits are highlighted. For these genes, the progress made in developing appropriate and valid rodent models to accelerate basic research is also detailed. Further, we discuss the recent development in the therapeutic management of epilepsy and provide a briefing on the challenges and caveats in identifying and testing species-specific epilepsy models.

## 1. Introduction

Epilepsy, a heterogeneous group of brain-related diseases distinguished by a lasting susceptibility to generating epileptic seizures, has continued to significantly burden society and families. Although initially termed a disorder, the International League Against Epilepsy (ILAE) redefined it as a disease to accentuate its significance and effect on the general population of patients and clinicians. An epileptic seizure, a hallmark feature of epilepsy disease, is defined conceptually as: “a transient occurrence of signs and/or symptoms” that are the clinical manifestation of “abnormal excessive or synchronous neuronal activity in the brain” [1]. Upon satisfying any of the following criteria, a clinical diagnosis of epilepsy disease can be given to a patient: (1) the patient encounters at least two unprovoked or reflex seizures that are separated by a duration of more than 24 h; or (2) the patient encounters one unprovoked or reflex seizure and shows a likelihood of suffering another seizure comparable to the general risk of recurrence of having two unprovoked seizures over the subsequent 10 years (i.e., a probability of more than 60%); or (3) the patient has an epilepsy syndrome (i.e., complex signs and symptoms unique to epilepsy conditions and (4) it occurs as an isolated event for a few seconds [2,3].

In 2017, the ILAE published an operational classification of seizures and epilepsies as an essential aid in diagnosing, treating, and understanding seizures and epilepsies, including the epilepsy incidence [4]. Classification of epilepsies combines information on seizure type, age of onset, etiology, clinical course, probability of remission, electroencephalographic (EEG) findings, radiologic findings, and genetics (Box 1) [4,5]. Generalized (distributed global brain networks) and partial or focal (localized brain networks) are the primary epilepsy types, along with idiopathic epilepsies of presumed genetic origin. Since the same mutations can produce different epilepsy types in other individuals, a single epilepsy type can be generated by mutations in more than one gene, leading to ambiguity in correlating current classifications with genetic causes [6].

Box 1Blood and Cerebrospinal Fluid Biomarkers for Epilepsy Diagnostics.Molecular biomarkers of seizures or epilepsy that promote fast, affordable detection have received significant attention in recent years. Purines such as adenosine triphosphate and adenosine are potent neuromodulators released during excessive neuronal activity that are altered during the developmental stages in epilepsy and act as a therapeutic target for the treatment of seizures and epilepsy [7]. Studies have shown that blood purine levels could be correlated with seizure severity and brain damage in mice, thus helping to identify patients with epilepsy [8].Glial cell activation and subsequent cytokine production following acute seizures have gained increasing support as important contributors to epileptogenesis. For example, interleukin-1beta (IL-1β), one of the most widely studied biomarkers for epileptogenesis, experiences an increased production following traumatic brain injury, thereby amplifying CNS hyperexcitability and excitotoxicity through Ca^2+^, glutamatergic, and GABAergic mechanisms, potentially culminating in epilepsy [9]. Markers of neural damage such as the neurofilament-1 (NFL) protein can distinguish between autoimmune phenomena and other types of epilepsy [10]. Programmed cell death protein 1 (PD-1) is critical in central and peripheral immunosuppressive mechanisms for regulating multiple signaling pathways that were found to be higher in epilepsy patients [11]. Clusterin (CLU), a glycoprotein, is involved in many biological processes, including tissue remodeling and differentiation, as well as cell proliferation and death. Human CSF-CLU levels were decreased in patients with both drug-resistant epilepsy and drug-responsive epilepsy compared to healthy individuals [12].Hence, there are biomarkers for diagnostic and prognostic purposes based on mutations that affect critical players in epilepsy and ASD co-occurrence. Nevertheless, a more robust analysis of neuronal firing patterns in the brain is suitable for understanding the features of these comorbidities.

Epilepsy comorbid with neurodevelopmental disorders (NDDs) such as autism spectrum disorder (ASD) and intellectual disability (ID) has been explained by underlying deficits in the excitation and inhibition balance in the widespread neuronal circuitry [13]. Epileptiform activity in a background of NDDs, particularly ASD, has been recapitulated in both mouse models [14] and human patients [15,16]. It is a challenge to tackle the co-occurrence of these two complex disorders; there have been several failures in clinical trials to develop actionable therapeutics. In this review, we present the common cause of epilepsy and the signature markers of epilepsy–ASD co-occurrence while focusing on mouse models and human patient data. In addition, the available therapeutic options and the challenges in bridging the gap between preclinical trials and behavioral phenotype amelioration are elucidated in order to present a holistic perspective on this unique comorbidity.

## 2. Causes of Epilepsy

Molecular components of neuronal signaling constitute almost all the genes identified thus far that cause idiopathic epilepsy, but the underlying genes lack clear inheritance patterns. The mutant alleles’ functional effects provide direct evidence that neuronal hyperexcitability is one cellular mechanism that underlies seizures [6]. In addition to these genetic causes of epilepsy, there are acquired epilepsies such as those elicited by traumatic injuries or other environmental contributors. These are detailed in the following sections, which primarily address those examples for which an epilepsy–ASD phenotype is well documented.

### 2.1. Genetic Causes of Epilepsy

Epilepsies may result from genetic abnormalities as primary causes or metabolic disorders as secondary causes, of which some also have genetic underpinnings. However, epilepsies associated with genetic abnormalities show considerable heterogeneity. Mutations in some genes such as *sodium voltage-gated channel alpha subunit 1* (*SCN1A*), can cause epilepsies or syndromes with epilepsy as the core symptom [17], while other genes such as *tuberous sclerosis complex-1&2* (*TSC1 and TSC2*) may be associated with gross brain developmental malformations and epilepsies [18] or neurodevelopmental disorders due to gene mutations such as Fragile X Syndrome (*FMR1*) [19], Tubulin β3 Class III, and WD Repeat Domain 62 (*TUBB3* and *WDR62)* [20,21].

The inheritance pattern of genetic epilepsies has commonly been classified into dominant (autosomal or X-linked) or recessive (autosomal or X-linked) inheritance [22]. However, epilepsy genes with undefined inheritance patterns, including genes with de novo mutations, have also been reported that are de novo associated with autosomal dominant genetic disorders as observed in *SCN1A* mutations, typically in Dravet syndrome [17] and in epileptic encephalopathies [23]. Additionally, there is also debate regarding whether the genetic contribution is largely from numerous common variants with a small effect discoverable by genome-wide association studies or from rare variants with a significant effect detectable by parallel sequencing studies [24].

#### 2.1.1. Ion Channel and Receptor-Mediated Causes

Approximately 25% of the genes that are implicated in epilepsy encode ion channels [24]. The voltage-gated sodium, potassium, and calcium channels have a common ancestral protein, as is evolutionarily evident from their shared domain characteristics. Mutations in these channels disrupt neuronal firings, and consequently, their functions, based on the data from human and mouse studies [3,6].

##### Voltage-Gated Sodium Ion (Na_V_) Channels

*SCN1A* which encodes the Na_V_1.1 subunit, is expressed predominantly in the GABAergic neurons and enriched at the axon initial segment, thereby regulating the initiation and propagation of action potentials in neurons [25,26]. *SCN1A* mutations are some of the most common epilepsy-related genes, with nearly a hundred mutations reported to date [25,27]. Heterozygous mutation in *SCN1B*, a β1-ancillary subunit of Na^+^ channels, has also been described in families with generalized epilepsy febrile plus [28,29]. The *β*-subunits of Na_V_ channels are enriched at axon initial segments and the nodes of Ranvier of both excitatory and inhibitory neurons, which regulates channel gating and expression; thus, improper modulation of neuronal functions results in epileptic seizures [30].

##### Voltage-Gated Potassium Ion (Kv) Channels

*KCNA1* encodes the K_v_1.1 subunit of the voltage-gated K^+^ channel (with α and β subunits), which is widely expressed throughout the central nervous system, especially in the hippocampus [31]. It is predominantly localized in the axon initial segment, axon preterminal, and the juxtaparanodal domain adjacent to the nodes of Ranvier, thus contributing to the regulation of membrane potential [32]. Human *KCNA1* heterozygous mutations are majorly associated with episodic ataxia type-1. *KCNA2* encodes the K_V_1.2 shaker-type voltage-gated K^+^ channel subunit, which is a part of the delayed rectifier class of K^+^ channels that is highly expressed in the central nervous system, particularly in the axon, and helps in the repolarization [33,34]. De novo mutations in *KCNA2* have been identified in cases of early infantile epileptic encephalopathy [35].

##### GABA Receptors

Genes that regulate synaptic transmission and DNA methylation/chromatin remodeling are often implicated in ASD and epilepsy. Exome sequencing of ASD cohorts identified several ASD gene candidates with de novo loss-of-function mutations or de novo missense variants such as *FMR1, TSC1*/*2*, and *SYNGAP1* [36], which, when mutated in rodent models, culminated in epilepsy and ASD [37]. Investigations of *DUP15q* syndrome, a specific form of “syndromic autism” marked by the co-occurrence of ASD and epilepsy phenotypes, have furnished direct genetic evidence of GABAergic gene defects associated with ASD–epilepsy comorbidity. *DUP15q* syndrome arises from maternal duplications on chromosome 15q11.2-q13.1, which leads to the overexpression of several genes, including *UBE3A* (ubiquitin ligase E3A) and a cluster of GABA_A_ receptor subunits, and phenotypically manifests in recurrent seizures (mainly infantile onset) [38]. The role of GABA receptors in this disorder has further been strengthened by association studies that linked single-nucleotide polymorphisms in GABA receptor subunit genes to ASD and the form of epilepsy [39].

##### Glutamate Receptors

Evidence from human genetics and mouse model studies has implicated excessive glutamate in ASD. However, increased levels of glutamate have been detected in both blood and platelets of subjects affected by ASD [40]. Most importantly, single-gene disorders characterized by ASD–epilepsy comorbidity (such as *FMR1*-, *TSC1/2*-, and *SHANK3*-mediated Phelan-McDermid syndromes) are caused by mutations in genes that regulate glutamate receptor-mediated signaling mechanisms.

#### 2.1.2. Metabolic Causes

There is a plethora of metabolic diseases associated with seizure occurrence that have many mechanisms: neurotoxic ammonia accumulation in urea cycle disorders, brain energy disbalance in glucose transport disorders, and impaired ATP production in mitochondrial disorders, to highlight a few [41]. However, in situations where the epilepsy phenotype co-manifests with ASD, there are distinct patterns. Metabolic disorders can be an integral direct underlying cause of GABAergic and glutamatergic transmission alterations. Chronically high levels of specific metabolites such as phenylalanine led to upregulation of glutamate receptors, with unwanted consequences on the E/I balance [42]. ASD, epilepsy, and ID comorbidity has also been characterized by amino acid metabolism dysfunctions such as inactivating mutations in branched-chain ketoacid dehydrogenase kinase (*BCKDK*) or amino acid transporter (*SlC7A5*) [43], altering brain amino acid profiles and causing neurobehavioral deficits in both humans and mice [43]. Other metabolic disorders associated with ASD and epilepsy include vitamin deficiencies (such as cobalamin and cerebral folate deficiency), mitochondrial diseases, and other disorders of energy metabolism; all of these conditions may be characterized by seizures and ASD [42,44,45].

### 2.2. Nongenetic Causes of Epilepsy

Environmental factors are contributors to the pathophysiology of ASD and associated comorbidities, which lack genetic underpinnings. Brain damage arising during delivery and through neonatal factors such as severe neonatal jaundice should be avoidable, but if treated inadequately, can predispose both to epilepsy and autism [46]. A more obvious environmental link is that of intrauterine infection. For example, maternal rubella during pregnancy has long been associated with a high risk of intellectual disability, autism, and epilepsy in the offspring [47]. Exposure to heavy metals such as mercury is associated with ASD and the occurrence of epilepsy. Although the mechanisms by which heavy metals may cause neurodevelopmental disorders have not been elucidated, the toxic effects of heavy metals on mitochondrial function, energy metabolism, and cell survival are known [42]. Several clinical studies have shown that exposure to the anticonvulsant and a mood stabilizer valproic acid (VPA) in utero is associated with an increased risk of ASD and epileptic seizures [48], which was corroborated by rodent studies of GABAergic dysfunction [49] and E/I imbalance [50]. Therefore, environmental factors further contribute to increased excitability in ASD.

## 3. Patient Studies of Epilepsy and ASD

A recent estimate by the Centers for Disease Control and Prevention (CDC) Autism and Developmental Disabilities Monitoring (ADDM) Network in the United States (US) reported a prevalence of ASD of about 1 in 44 children who typically show two peaks in epilepsy incidence: one in early development and the other in teenage years [51]. A 2017 meta-analysis revealed a prevalence of 6.3% of epilepsy in patients with ASD, and with the risk being further compounded by the presence of intellectual disability (ID), younger populations (<18 years) were found to be more susceptible [51]. The characteristics of epilepsy in ASD are categorized based on the onset of a seizure, seizure types, EEG findings, and epilepsy outcome. A study on the developmental evaluation of more than 100 ASD-afflicted individuals showed an incidence of epileptic seizures in a quarter of the patients, with age at onset varying from 8 to 26 years [52].

### 3.1. Mechanisms of Epilepsy–ASD Comorbidity

Epilepsy–ASD comorbidity can be explained by a common hypothesis that postulates that neurodevelopmental deficit of multiple origins (e.g., genetic, metabolic, immune, and environmental) results in an altered structure and function of excitatory and inhibitory circuits. A persistent E/I imbalance and hyperexcitability are caused by aberrant neuronal activity. Neurodevelopmental deficits in inhibitory circuits and the subsequent E/I imbalance are mainly due to defects in GABA-mediated activities and hyperexcitability caused by the increased glutamatergic signaling and function [13]. During embryonic and early postnatal brain development, all these deficits may hamper synaptic plasticity and neuronal connectivity and can manifest in hyperexcitability; cognitive, social, and emotional deficits; and intellectual disability (ID) [53].

Other than the simplistic view of E/I imbalance as the mechanism behind the co-occurrence of ASD and epilepsy, many genetic mutations cause ASD, epilepsy, or both. Structural proteins that anchor synaptic machinery, regulate synaptic vesicle release, and govern the migration of neurons and organization of network connections are associated with ASD and epilepsy [14].

### 3.2. Hallmark EEG Signatures in Epilepsy–ASD Comorbidity

Electrophysiological-related measurements are an essential tool to capture seizures in human patients with epilepsy and in animal-model-based studies. To precisely locate brain areas where the seizure originates, electroencephalography recordings from the scalp or sometimes directly from the brain are used that further help in subsequent resection during surgical intervention procedures [54]. A characteristic feature of EEG recordings in individuals with epilepsy besides ictal (neuronal firing during a seizure between seizure onset and at the end) occurrences include transient electrophysiological disturbances that arise between ictal episodes more frequently than seizures, such as interictal EEG spikes and sharp waves (IIS), pathological high-frequency oscillations (HFOs) between 80 and 600 Hz [55], and “micro seizures” and “micro-periodic epileptiform discharges” [56,57].

A noticeable increase in the incidence of epileptiform activity (interictal spikes) in electroencephalogram (EEG) recordings has been reported in ASD-afflicted human patients [15,16]. Altered γ-band oscillations have also been described in resting-state EEGs from ASD patients [58], suggesting dysfunction of parvalbumin (PV)-positive interneurons (which are responsible for these band oscillations). However, the results of in-depth analyses of the EEG profiles of patients who had ASD and epilepsy compared with only ASD are still unclear (Table 1). In the following sections, we have highlighted the epilepsy characteristics for a few of the key genes implicated in different types of ASD by citing data gleaned from patient cohort studies.

#### 3.2.1. *MeCP2*

Mutations in the gene methyl-CpG-binding protein (*MeCP2*) cause Rett syndrome, a rare and severe genetic encephalopathy that affects roughly 1:10,000 females born each year [69]. Rett syndrome (RTT) is a clinically defined syndrome that includes developmental regression followed by stabilization, partial or complete loss of purposeful hand skills and spoken language, gait abnormalities, epilepsy, and stereotypic hand movements [70,71,72]. Although one study reported an improvement of epilepsy in adult age in RTT-afflicted individuals [73], other studies claimed that epilepsy persists as a major concern even in adulthood [74,75]. Typically, three distinct seizure patterns are seen in RTT: (a) no seizures, (b) frequent remissions and relapses, and (c) unremitting and persistent seizures [76]. Pathogenic mutations in *MECP2* (methyl-CpG binding protein 2) may be accompanied by active epilepsy (the seizures last five years), but the exact relationship of epilepsy with the mutation type remains unclear [69]. Patients with RTT are classified into the early seizure (Hanefeld) variant and the congenital (Rolando) variant; these classes may have cyclin-dependent kinase-like 5 (CDKL5) [70,77] or fork-head box G1 (FOXG1) mutations [70,78,79], respectively [80,81,82], suggesting that patients with features of RTT without *MECP2* mutations may have mutations in other genes, including those associated with epileptic encephalopathies, intellectual disability (ID), or autism spectrum disorder [83].

#### 3.2.2. *SYNGAP1*

*SYNGAP1* encodes for a RAS-GTPase activating protein—SYNGAP1—that is expressed highly in the brain and is involved in the modulation of excitatory synaptic transmission by N-methyl-D-aspartate (NMDA) receptors [84]. Pathogenic variants of the protein, primarily truncation mutant products, have been reported in individuals with intellectual disability (ID) and ASD with or without seizures [85,86]. *SYNGAP1* heterozygous mutations disrupt the E/I balance in the developing brain, resulting in accelerated glutamatergic synapse maturation and altered synaptic plasticity alongside cognitive, emotional, and social deficits [87,88]. A distinct developmental and epileptic encephalopathy (DEE), termed SYNGAP1-DEE, has been defined as psychomotor delay preceding epilepsy onset, with seizures appearing at a mean age of two years [62,85,89,90]. Individuals present mostly generalized seizures, namely: myoclonic, atonic, and myoclonic-atonic seizures; atypical absences; eyelid myoclonia; and myoclonic absences [86,87,88].

Electroclinical seizure patterns of 15 (10 females, 5 males) previously unreported individuals with *SYNGAP1* pathogenic variants (de novo missense or truncating mutations) revealed atypical absences in 8 individuals associated with atonic phenomena, oculoclonic movements, or rhythmic axial myoclonia, in addition to other seizure patterns [91]. Myoclonic seizures, eyelid myoclonia with absences (EMA), atonic seizures, myoclonic absences, atonic seizures, and myoclonic-atonic-tonic seizures were common patterns. Interestingly, three of the patients presented with the same mutation but exhibited various epilepsy phenotypes for ID and ASD, reflecting the diversity and unpredictability of epileptic phenomena even under the umbrella of the same NDD [91]. This results of this study strongly correlated with the visual cortex hyperexcitability recently reported in mouse models with *Syngap1* haploinsufficiency [92]. Another study reported a similar pattern of epileptic seizures in patients with *SYNGAP1* mutations [62].

#### 3.2.3. *FMR1*

Fragile X syndrome (FXS), the most prevalent X-linked monogenic cause of ID/ASD, is characterized by homozygous null *FMR1* mutations and epileptic seizures [93,94]. Data from the Fragile X Online Registry With Accessible Research Database, a multisite observational study initiated in 2012 involving FXS clinics in the Fragile X Clinic and Research Consortium, reported that out of 1607 participants, 12% of patients with a 77% male prevalence had at least one seizure incidence, with this rate being significantly higher in males than females. Interestingly, individuals with FXS without seizures were less prone than those displaying seizures to have ASD/ID, sleep apnea, language deficits, and other related behavioral deficits. Most of the cohort had an onset of seizures of more than 10 years prior to the study. However, patients who had seizures for more than 3 years suffered from greater cognitive and language impairment, but not behavioral disruptions, compared with those with seizures for <3 years [94]. A developmental evaluation of EEG findings in FXS patients from 2 to 51 years of age showed centrotemporal spikes as the most common epileptiform pattern [63].

Seizures can be readily controlled for most individuals with FXS and tend to disappear in adolescence [95]. A voltage-gated inward current, *I*_mGluR(V)_, has been proven to be the cellular basis for the epileptogenic behavior induced by activation of the mGluR_5_ receptor. Stimulation of mGluR_5_ by the agonist dihydroxyphenylglycine in mouse hippocampal slices caused prolonged epileptiform discharges that lasted for more than 1 h after washout of the agonist [96], which may explain the mechanisms of neuronal dysfunction in FXS that might underlie hyperexcitability leading to seizures in this disorder.

#### 3.2.4. *SHANK1-3*

The SH3 and multiple ankyrin repeat domains (*SHANK*s) are a family of scaffolding proteins in excitatory synapses required for synaptic development and function. *SHANK3* is the most recognized member of the family, being associated with Phelan-McDermid syndrome [97]. Molecular defects of *SHANK3 underlie* several neurodevelopmental entities, in particular ASD and epilepsy, whereas there is a paucity of data on the disease associations of *SHANK1* and *SHANK2*. One study reported a novel de novo mosaic *p.*(*Gly126Arg*) *SHANK1* variant as the monogenic cause of disease in an 11-year-old male patient who presented, from the age of 2 years, moderate ASD/ID and refractory epilepsy of the Lennox-Gastaut type [67], which is an epileptic encephalopathy that is refractory to treatment and is characterized by three major features: polymorphic seizures, intellectual impairment, and a characteristic electroencephalogram (EEG) pattern [98]. In silico analyses revealed that the *SHANK1* variant disrupted a conserved region of SHANK1 with high homology to a recently recognized functionally relevant domain in SHANK3 that is implicated in ligand binding, including the “non-canonical” binding of *RAP1* [67].

The spectrum of seizure semiologies and electroencephalography (EEG) abnormalities has not been studied in detail in the context of *SHANK3* mutations. Microdeletions or heterozygous loss-of-function mutations at the *SHANK3* locus reveal a wide spectrum of seizure semiologies and frequencies. Electroencephalographic abnormalities are heterogenous: from slowing or absence of the dominant occipital rhythm to focal spike and slow-wave discharges to generalized spike and slow-wave discharges [99]. One previous study investigating the seizure types associated with Phelan-McDermid syndrome suggested that epilepsy associated with this disorder is mild and pharmacologically controllable [100]. However, there are no consistent neuroimaging or migration-related abnormalities that are likely to predispose *SHANK3* patients to seizures [99].

#### 3.2.5. *TSC1*

The tuberous sclerosis (TSC)/mTORC1 signaling pathway is a major PI3K/PTEN/AKT downstream pathway that mediates cellular and behavioral effects in the nervous system [68]. Tuberous sclerosis complex (*TSC*), a congenital syndrome characterized by the widespread development of benign tumors in multiple organs, is caused by heterozygous mutations in one of the tumor-suppressor genes (*TSC1* or *TSC2)*. About 80% of affected patients have de novo mutations, with the remaining 20% having inherited *TSC* gene mutations. Epilepsy, mental retardation, and ASD/ID comprise the most common neurological manifestations of *TSC* mutations [68].

The prevalence of *TSC* mutations that cause ASD in the population is 1–4%, whereas features of ASD are present in 25–50% of individuals with *TSC* mutations [100,101]. A nonspecific disruption of brain function due to *TSC*, including tuber location or seizures and their effects on brain development, may be underlying reasons for this strong association. In children with *TSC* mutations with ASD, seizure onset occurs at significantly earlier ages than in children without ASD [68]. Epilepsy is the most common neurological symptom in *TSC,* with 60–90% of such patients developing epilepsy during their lifetime [102]. Furthermore, the most common seizure types reported in patients with *TSC* mutation are complex partial, generalized tonic-clonic, myoclonic, and infantile spasms. The epilepsy is often quite severe; poor prognostic signs include multiple seizure types, seizure onset before one year of age, and multifocal EEG abnormalities [68]. The multifocal occurrence of cortical tubers substantiates the occurrence of the multifocal nature of epileptic foci in *TSC* mutations. In addition to electroencephalography, epileptogenic areas can be detected using magnetic resonance imaging (MRI) and positron emission tomography (PET) scans [103,104].

Although seizures may play a causal role in developing encephalopathy, it is uncertain whether epilepsy in *TSC* mutations is simply a marker in infants prone to developing encephalopathy. Many factors influence the presentation of both early seizure onset and encephalopathy, resulting in neurodevelopmental deficits. Mammalian target of rapamycin (mTOR) overactivation caused by *TSC* mutations leads to altered cellular morphology with cytomegalic neurons, dysregulated synaptogenesis, and an imbalance between excitation/inhibition, thus providing a likely neuroanatomical substrate for the early appearance of refractory seizures and the encephalopathic process [105]. In addition to these molecular changes, EEG data collected from patients with *TSC* mutations and without ASD and from patients with non-syndromic and nonclinical controls revealed an altered neuronal network topology. These data represented a functional correlate of structural abnormalities that may play a role in the pathogenesis of neurological deficits [106].

Despite patient-cohort-based studies in human NDDs (Figure 1), a gap remains in the understanding of the underlying mechanisms in humans due to ethical constraints, necessitating the need to use other mammalian models. In addition to the reported occurrences of epileptic phenotypes in human neurodevelopmental disorders, mouse models specific to epilepsy have been generated to dissect the mechanisms of this heterogeneous disease. Epilepsy features have also been extensively studied in mouse models created to study neurodevelopmental disorders. These will be discussed in the following sections.

## 4. Animal Models of Epilepsy and NDDs

Animal models are crucial to studying any physiological condition or disorder and its respective pathophysiology due to the ethical restrictions on human studies. Mouse and rat are among the most preferred species for biomedical research due to their anatomical, physiological, and genetic similarities to humans [107]. Regarding the animal models in epilepsy, priority is given to chronic epilepsy models capable of exhibiting epileptic behaviors such as status epilepticus (SE), a condition with continuous seizures lasting more than 30 min or two or more seizures without full recovery of consciousness during this episode [108,109]. A febrile seizure, another type of seizure that can be observed in the animal models of chronic epilepsy, occurs during fevers accompanied by loss of consciousness (>38.0 °C/100.4 °F) and is caused by an infection in children between 6 months and 5 years old. Studying these models provides an insight into the pathophysiological processes involved in human epilepsy [110,111,112,113]. Animal models of epilepsy can be created by either genetic manipulation or induction at a postnatal stage via chemoconvulsants or electrical stimulations.

### 4.1. Nongenetic Models of Epilepsy

#### 4.1.1. Chemical Convulsant

Pilocarpine and kainic acid are two primary compounds whose local or systemic administration causes limbic seizures and SE. Pilocarpine is a muscarinic acetylcholine receptor agonist that results in limbic SE as well as cognitive and memory deficits [114,115,116]. One method of administering pilocarpine in rats is to induce SE with a low mortality rate of 3 mEq/kg dosage intraperitoneally after lithium chloride pretreatment for 24 h [116,117,118,119]. Studies using an administration of high (400 mg/kg) and low (100 mg/kg) doses of pilocarpine showed a sequence of behavioral alterations that eventually led to limbic status epilepticus in the case of increased pilocarpine administration, while a similar seizure activity was exhibited with a different threshold for the lower dose [115]. This compound is thus capable of inducing widespread brain damage, as the epileptiform activity starts from the hippocampus and then migrates to the amygdala, cortex, and subsequently both limbic and cortical leads [115,120]. There are two main patterns of seizure onset for medial temporal lobe epilepsies (MTLE) induced by pilocarpine: the hypersynchronous (HYP) onset pattern and the low-voltage fast (LVF) pattern [121,122,123]. Seizures usually have a beginning (prodrome and aura), middle (ictal), and end (postictal) stage. The HYP pattern is characterized by periodic focal (pre-ictal) spiking at approximately 2 Hz and a following high-amplitude, low-frequency ictal activity in the hippocampus and subiculum [122,124,125,126]; whereas an initial positive or negative spike followed by low-amplitude, high-frequency activity is noted as the LVF pattern [122]. Kainic acid (KA) is an L-glutamate analogue that leads to neuronal depolarization and seizures primarily in the hippocampus when administered intracerebrally (0.1–3.0 mg per hemisphere) or systemically (15–30 mg/kg) [127,128]. Later, another protocol that demonstrated a relatively lower mortality rate was devised that used intraperitoneal administration of lower doses (5 mg/kg) [129]. Other studies reported that the high expression of GluK2 receptors at the mossy fiber input synapses and strong recurrent connections rendered the CA3 region of the hippocampus essential for the induction of KA-mediated seizures [130,131,132,133]. Hence, pilocarpine and kainic acid have been used to generate animal models of spontaneous recurrent seizures [134], but they result primarily in temporal lobe epilepsy (TLE) [135].

PTZ-induced kindling is another well-accepted model of chronic epilepsy [136] that is capable of inducing SE. However, the seizure characteristics have been observed to vary with different postnatal developmental stages [137,138,139]. Generalized tonic-clonic seizures in rats can be induced by administration of PTZ (20 mg/kg) every 48 h [140], while 50 mg/kg of PTZ administration every 24 or 48 h achieved kindling in 80% of mice after 15 injections [141]. PTZ kindling is a time-saving method to induce seizures consistently with comparatively fewer mortality rates [136,142,143]. Electroencephalography (EEG) recordings of PTZ-induced seizures showed tonic-clonic seizures with high amplitude spikes, polyspikes, and sharp spike-wave discharges and a depression/suppression of the EEG after tonic-clonic convulsions, similar to that seen in epileptic patients [144,145].

Another potent stimulant that causes epileptic seizures is flurothyl, a volatile compound that has been shown to cause myoclonic jerks, forelimb clonus, wild running, and tonic posturing followed by a recovery period [139]. The volatile property of flurothyl makes it feasible to administer the compound via inhalation and facilitates the determination of the seizure threshold of the animals [146]. It has been reported that flurothyl inhalation results in irreparable neural damage in areas such as the cerebral cortex, hippocampus, amygdala, thalamus, basal ganglia, and mesencephalon [147]. Certain studies have also shown that when recurrent seizures are induced during the early stages of life in rodents by exposing them to flurothyl, the seizure susceptibility increases when they become adults [148,149]. Flurothyl-mediated seizure inductions were also found to be as clinically effective as electrical inductions with lesser effects on memory functions. In the case of electrical inductions, the path and the dose of the current and the seizure itself leave a mark on the brain, as seen in psychological tests and the EEG, whereas the effects of flurothyl on the brain are those of the seizure alone [150]. Moreover, exposing mice to flurothyl-induced generalized seizures (lasting for <30 s) for 8 days caused 95% of the mice to have spontaneous seizures after the treatment period, which lasted around 4 weeks [151]. The model from this study may provide insight into why spontaneous seizures remit without anticonvulsant treatment.

#### 4.1.2. Electrical Stimulation

Seizures can also be induced in animal models using electrical stimulation. It is considered to be less harmful than the chemoconvulsants, which have higher mortality rates and high variability in the frequency and severity of spontaneous seizures [135]. Electroshock-induced seizures (ES), after discharges (AD), and kindling are the different electrical stimulation techniques employed to induce seizures in the target animal models. ES involves whole-brain stimulation; for example, 6 Hz in mice and 50–60 Hz in rats, and it is categorized into minimal ES with minimal clonic behavioral seizures mainly within the forebrain [135,152] and maximal ES displaying generalized tonic-clonic seizures occurring in the hindbrain [153]. Unlike ES, AD is a focused approach with induction in specific brain regions such as the hippocampus, where seizures are observed following the postictal refractoriness [154]. The animals display complex partial seizures if AD is applied to limbic structures and myoclonic seizures if applied to the sensorimotor cortex. When AD is repeatedly induced in a specific brain region such that it enhances the seizure susceptibility progressively with each AD leading to a permanent epileptic state with spontaneous seizures, it becomes a kindling model of ES [155,156,157]. Temporal lobe epilepsy is modeled based on the electrical kindling of a limbic structure. Hippocampal kindling with a train of stimuli (≥80) of 60 Hz for 2 s results in spontaneous recurrent seizure (SRS) events, which are distinguished by EEG discharges and associated motor seizures [158]. However, the process for ES, especially kindling, is expensive and extensive, as it involves extended periods of handling and stimulation procedures [139].

#### 4.1.3. Traumatic Brain Injury

Mild to moderate brain injuries can lead to complications of seizures; in some cases, the spontaneous reoccurrence of seizures develops into post-traumatic epilepsy (PTE). Animal models of PTE are essential for understanding the pathophysiology of the resultant epileptic seizures, as the condition is prone to pharmacoresistance, increasing the necessity for effective therapeutic strategies [159,160]. Some of the existing animal models of PTE include the fluid percussion injury model, controlled cortical impact (CCI) model, impact acceleration model, canine model of post-traumatic epilepsy, penetrating head trauma model, and pediatric post-traumatic epilepsy model, which replicate the neuroinflammatory, metabolic, and neurodegenerative characteristics of PTE patients [161]. Among these, the most extensively studied model is the fluid percussion injury model [162,163], which was developed by applying pressure pulses of 0.9–2.1 atm (1.5–3.4 atm for rats) for approximately 20 ms after performing craniotomy over the right parietal cortex while keeping the dura intact in mice [164,165]. CCI models of rats and mice are also important models for PTE that are created by causing cortical malformations in the exposed brain by utilizing pneumatic or electromagnetic impactors at different velocities for varying severity [166,167]. Ablation of brain injury centered over the left or the right sensorimotor cortex created asymmetrical responses [168]. Another model involved the administration of homocysteine (845 mg/kg intraperitoneal) 16–18 h after cobalt implantation, resulting in refractory cortical-onset SE accompanied by injury, with evidence of widespread neocortical oedema and damage. This was similar to the conditions observed in SE arising from traumatic brain injury, subarachnoid hemorrhage, and lobar hemorrhage [169,170]. A two-hit model for epilepsy was also developed by utilizing optogenetic kindling in which a secondary factor of injury and inflammation could be added to induce a spontaneous seizure, allowing for the investigation of injury or the role of inflammation in epileptogenesis without being hindered by electrode insertion, as in the case of electrical kindling [171,172].

### 4.2. Genetic Models of Epilepsy

Animal models with their genetics as the root cause for seizures rather than chemical compounds or voltage induction have also been developed to examine the various aspects governing epilepsy. One of the well-studied genetic models of epilepsy is the mouse model susceptible to an audiogenic seizure (AS), which is characterized by violent generalized seizures upon loud or intense auditory stimulation [173]. The inferior colliculus in the auditory midbrain is the primary structure involved in an AS [174,175]. The *DBA/2* inbred strain of *Mus musculus* is an example that displays convulsions that are fatal on exposure to even doorbells (~10–120 kHz, 90–120 dB) [176]. A few other strains of rats that show audiogenic seizures include the Krushinsky–Molodkina (KM) strain of Wistar descent [173], the University of Arizona (UAZ) strain of Sprague Dawley descent [177], the genetically epilepsy-prone rat in the United States [178], P77PMC rats [179], Wistar Albino Glaxo/Rijswijk rats (WAG/Rij) [180], the Wistar Audiogenic Sensitive rat [181], and the Wistar Audiogenic rat (WAR) [135,182]. The seizures exhibited by these animals are strain-specific but mainly follow a sequence of actions that initiate with a startle response, then a momentary quiescence followed by violent running, tonic-clinic seizure, and a postictal depression phase [135,183,184]. Extended exposure of the animals to AS protocol is termed audiogenic kindling (AuK), which can sometimes result in the development of limbic seizures during which new behaviors such as facial and forelimb clonus followed by elevation and falling are observed [181,185,186]. Even though these animals need a trigger for the induction of epileptic activity, an advantage of these models is that the stimulus specificity helps to avoid random seizures and reduce mortality in the animals [135].

Rats with petit mal or absence seizures such as the Genetic Absence Epilepsy Rat from Strasbourg (GAERS) and WAG/Rij exhibit spontaneous bursts of 7 to 11/s and 200 to 400 μV in amplitude with a duration of 0.5 to 40 s that occur hundreds of times a day. These seizures persist throughout the rats’ lifetime, and these animals exhibit behavioral arrest and frequent facial myoclonia [187,188,189]. The different mouse models of absence seizure include those exhibiting lethargic, tottering, leaner, and stargazer phenotypes caused by monogenic mutations in the genes for murine voltage-gated Ca^2+^ channels [190]. For example, the tottering mutation (*Cacna1atg*) that occurs in the *Cacna1a* for the CaV2.1 subunit gives rise to polyspike discharges and behavioral absence seizures [190,191,192], while the leaner mutation in the same gene leads to cortical spike-wave discharges in the animals with cerebellar atrophy [190,191,193].

Apart from these, there are many validated genetic mouse models of epilepsy. Some mouse models, including *Arfgef1*, *Fmr1*, *Pcdh19*, *Syngap1*, and *Ube3a*, do not exhibit spontaneous seizures, although when observing the EEGs, with the exclusion of *Pcdh19*, all the others show increased susceptibility to induced seizures. Some of these models display spontaneous seizures in their early life, such as those with homozygous truncation mutation in the *Gabra1* gene, which develops a severe seizure phenotype by postnatal day 19 [194], while some exhibit the spontaneous seizure phenotype in the later phases of their life (~PND300), as observed in a null or missense mutation in *Cdkl5* [195]; certain other mutations cause sudden death, such as a homozygous null mutation of *Scn1b* gene [196,197]. Below, we discuss some of the genetic mouse models of neurodevelopmental disorders in which the mutations have been implicated in epileptic seizures as well (Figure 2).

#### 4.2.1. *SCN1A*

Allelic variants of sodium voltage-gated channel alpha subunit 1 (*SCN1A*) are closely associated with generalized epilepsy that includes febrile seizures and epileptic encephalopathy [198]. The mutations in *SCN1A* result in a wide range of epilepsies that differ in their comorbidities and functional deficits, such as Dravet syndrome (DS), genetic epilepsy with febrile seizures plus (GEFS+), and developmental and epileptic encephalopathies (DEEs), which further comprises myoclonic-atonic epilepsy (MAE) and epilepsy of infancy with migrating focal seizures (EIMFS) [198,199]. Furthermore, several studies investigated the autistic behavior displayed by mice with mutated *Scn1a,* making them relevant to the studies linking ASD and epileptic seizures [200,201,202].

The heterozygous knockout (KO) *Scn1a*^+^^/^^−^ mouse model with Dravet syndrome (DS) showed a reduced level of NaV1.1 to 50% of the normal levels and spontaneous seizures, hypothermia-induced seizures, and a high mortality rate within one month of birth while the homozygous KOs of the same mutation died within 15 days of birth [26,203]. Their seizures usually lasted for 20 s and included both clonic and tonic-clonic seizures. The disruption of the NaV1.1 channels resulted in ataxia and related functional deficits that impacted the GABA function as observed in the GABAergic cerebellar Purkinje neurons, [204]. Whole-cell voltage-clamp recordings were made using different *Scn1a*^+^^/^^−^mice bred to different strains such as 129S6/SvEvTac (129.*Scn1a*^+^^/^^−^), which show normal survivability, and C57BL/6J (F1.*Scn1a*^+^^/^^−^), which show premature lethality. The recordings displayed a decrease in the sodium current (INa) density in GABAergic interneurons in P21 F1.*Scn1a*^+^^/^^−^ and an elevated I_Na_ in pyramidal neurons in both strains, suggesting that the difference in pyramidal neuron excitability due to altered I_Na_ density may lead to strain-dependent seizure severity and survival [205].

EEG recordings from the *Scn1a*^+^^/^^−^ model showed normal periods of low-amplitude baseline cortical activity, spontaneous electrographic and/or behavioral seizures, and epileptiform interictal activity [26]. A power spectral analysis of the background EEG activity from *Scn1a*^A1783V^ revealed normal background oscillations at the pre-epileptic stage, a marked reduction in the total power during the onset of severe epilepsy, and a subsequent smaller reduction later in life. They also noted that a low EEG power at the stage of severe, frequent convulsive seizures correlated with an increased risk of premature death [206]. Furthermore, calcium imaging from acute brain slices indicated a significant dysfunction in the filtering of the perforant path input to the dentate gyrus (DG) in young adult *Scn1a*^+^^/^^−^ mice as a result of enhanced excitatory input to DG neurons, which established that the cortico-hippocampal circuit is a key locus of the pathology in *Scn1a*^+^^/^^−^mice [207]. However, the mean activity of the cortical PV-INs was increased in the mouse model, which could have contributed to the ictal/middle phase [208]. In another study in which current-clamp recordings were obtained from PV-INs of mice with a heterozygous *K1270T* (KT) GEFS+ mutation in the *Scn1a* gene (*Scn1aKT*), the mice exhibited heat-induced seizures when exposed to a temperature of ~42 °C, which revealed that the mutation caused an increased threshold and a decreased amplitude for the action potential without altering the intrinsic membrane properties [209,210]. Studies have shown that the alteration of gene expression in rodent models of Dravet syndrome can result in seizure reduction [211]. Moreover, among all the animal models of *Scn1a* mutation, the Dravet models have displayed successful face validity, making them suitable for testing novel therapeutics that are capable of alleviating the seizures along with other setbacks of the mutation and taking treatments one step closer to success.

#### 4.2.2. *Syngap1*

Synaptic RAS-GTPase activating protein 1 (SYNGAP1) is a major signaling protein found mainly in the excitatory synapses that plays a pivotal role in regulating fundamental molecular changes in dendritic spine synaptic morphological and functional modifications [87,88,212]. De novo mutations in *SYNGAP1 cause* intellectual disability (ID), ASD, and epilepsy [85,89,90]. The excitation/inhibition (E/I) balance is an essential factor that modulates cognitive and other functions [213,214]; an imbalance can cause epilepsy, as observed in neurodevelopmental diseases such as ASD.

Studies have shown that *Syngap1^+^*^/^^−^ mice had a reduced fluorophenyl-induced seizure threshold and were susceptible to audiogenic seizures [86,215]. Cortical EEG recordings from *Syngap1^+^*^/^^−^ mice displayed generalized sharp epileptiform discharges and occasional brief (<1 s) or prolonged (>10 s) seizures with a myoclonic jerk [216]. Continuous 24-hour subdural vEEG/suprascapular EMG recordings from the cortex at temporally advancing ages identified spontaneous seizures in 50% of *Syngap1^+^*^/^^−^ mice [217]. The seizures observed were mainly myoclonic at postnatal day (PND) 60, began during non-rapid eye movement (NREM) at transitions from NREM to waking, and lasted for around 30–40 s with rhythmic spike-wave discharges occurring at ~3–4 Hz [217]. After PND 120, the mutant mice were found to exhibit multiple seizure phenotypes, including myoclonic (NREM), generalized tonic-clonic (NREM), and electrographic seizures (wake); these were consistent with the clinal reports [62,214,218]. Interictal spikes (IISs) were observed during both sleep and waking in the *Syngap1^+^*^/^^−^ mice, with an increase in the frequency during NREM of the mice in the P120 age group [217]. However, it has been shown that the re-expression of *Syngap1* or pharmacological administration in *Syngap1^+^*^/^^−^ adult mice showed improvement in the seizure threshold [86,219]. Genetic restoration of the *Syngap1,* and subsequently the protein level*,* restored the threshold level to that of healthy mice [216,220], implying a potential therapeutic target.

#### 4.2.3. *Fmr1*

Fragile X syndrome (FXS), which is the most common cause of inherited mental retardation, arises due to mutations occurring in the *FMR1*. Several mouse models have been generated to study this condition, which exhibits phenotypes analogous to the clinical and pathological symptoms observed in human patients [221]. Unlike fragile X patients who exhibit spontaneous seizures [222], fragile X knockout mice do not exhibit spontaneous seizures, but only when they are exposed to intense stimulations such as auditory [223,224]. The KO models were constructed by disrupting exon 5 of the *Fmr1* gene using a neomycin gene, and they showed an increased susceptibility to audiogenic seizures at all the ages tested compared to their WT littermates [224,225,226]. These studies showed that exposing mice at PNDs 17, 22, 35, and 45 to an electric doorbell at 120 dB for 60 s induced a sequential seizure response in them that consisted of an early wild running phase followed by generalized myoclonus and tonic flexion and extension, and sometimes followed by respiratory arrest.

EEG studies confirmed the hypersensitivity observed in *Fmr1* KO mice, which displayed an increased evoked EEG gamma power (30–80 Hz). It was normalized in the treatment of the mice with racemic baclofen, a GABA_B_ agonist that improved the working memory of *Fmr1* mice [227], suggesting a potential therapeutic target. Another EEG study indicated a sleep-enhanced interictal epileptiform discharge that appeared as “rolandic” spikes in the centrotemporal regions during the non-REM sleep [63]. Paired whole-cell recordings from pyramidal neurons in *Fmr1* KO mice revealed reductions in synchronized synaptic inhibition and coordinated spike synchrony in response to the group I metabotropic glutamate receptor (mGluRs) agonist 3,5-dihydroxyphenylalanine (DHPG), implying a weakened somatostatin-expressing, low-threshold-spiking (LTS) interneuron network in layer II/III of the somatosensory cortex, resulting in altered activity of the cortical network that was in line with the FXS phenotype [228].

Spontaneous neuronal ensemble activity recorded during sleep in the somatosensory cortex of *Fmr1*^−^^/^^−^ mice indicated an abnormally high synchrony of neocortical network activity and a threefold increase in the neuronal firing rates, showing that the cortical networks in FXS are hyperexcitable in a brain-state-dependent manner, which explains the several dysfunctions associated with FXS, including intellect, sleep, and sensory integration [229]. The phenotypes of this genetic error are similar for humans and animal models [226,230,231], suggesting conserved sensory processing circuits [232] and allowing us to further explore the underlying mechanisms of FXS and the accompanying seizures.

#### 4.2.4. *Shank3*

Mutations in *Shank3* have been implicated in the functional integrity of dendritic spines, and therefore *SHANK3* haploinsufficiency can cause epilepsy risk due to abnormalities in the glutamatergic synaptic structure and function [233]. However, most of the *Shank3* knockdown mouse models do not display seizure phenotypes [234,235]. The *Shank3b*^−^^/^^−^ mice do not display any spontaneous seizures, including PTZ-induced seizures, but they may exhibit seizures on rare occasions such as during handling in routine husbandry procedures [234,236]. Another mouse model, *Shank3^Q321R/Q321R^*, lacked spontaneous seizures but had a decreased δ-band concomitant with an increased α-band in EEG in the frontal and parietal lobe, respectively, and reduced neuronal excitability in the hippocampal CA1 neurons [237]. This was further confirmed by another study in which hyperexcitability discharges; electrographic seizures in CA1, DG, and the frontal cortex; and an increased frequency of epileptiform spikes in the DG were observed [238]. These studies validated the fact that epileptic seizures can be due to the synaptic E/I imbalance observed [238,239,240].

Chemoconvulsants, electrical stimulations, and genetic manipulations are some of the key methods used to develop animal models of epilepsy. Each method has its advantages and disadvantages, making it difficult for scientists to exactly map intricate details of different types of epileptic seizures seen in humans. However, the existing techniques do let researchers explore specific features of each epileptic condition and draw conclusions on the possible underlying causes, which may help to develop new treatment strategies. In addition to mouse models, studies have also been done to explore epilepsy characteristics in induced pluripotent stem cells from human patients, as outlined in the following sections.

## 5. Current Therapeutic Management and Antiepileptic Drugs in Clinical Trials

The currently available antiepileptic drugs (AEDs) modulate the voltage-gated channels, GABAergic and glutamatergic transmission, and newly emerged players such as carbonic anhydrase and synaptic vesicle proteins [241,242,243]. However, these druggable targets provide symptomatic relief (suppression of seizures) without employing preventative and curative effects. Additionally, 35% of the patients did not respond to the monotherapies or polytherapy of available AEDs [244]. Developmental epilepsies constitute the major portion of pharmacoresistant seizures. A detailed understanding of initiation- and progression-phase biomarkers and targeting of the underlying pathogenesis will provide a critical space to unravel the novel targets and maintenance of neuronal homeostasis [245,246]. In line with this, several novel targets and processes have been shown to elicit beneficial effects on the excitation/inhibition balance in animal models [243]. Clinical studies on the repurposing of anticancerous, antimicrobial and anti-inflammatory drugs for antiepileptic potential highlighted the importance of the involvement of novel targets in overlapping pathways, which contribute to the severity, frequency, and intensity of epileptic seizures [247,248]. However, the mechanisms for major drugs in clinical trials still range from the old treatment strategies such as the modulation of channels to inhibition of GABAergic transmission (Table 2 and Table 3). The new generation of antiepileptic strategies should involve timely intervention with double-edged efficacy in targeting the symptomatic as well as antiepileptogenic and/or disease-modification (from pathological to physiological state) targets. This reflects the need for stronger contenders to impede the underlying molecular pathways of epilepsy and for an improvement in current modeling approaches that can closely mimic the pathology, etiology, and unpredictability of human seizures. In the next two sections, we review the species-specific differences as a major limitation of rodent models and recent advancements in modeling approaches to tackle these challenges, respectively.

### 5.1. Species-Specific Differences between Rodent and Human-Derived Models of Epilepsy

Pharmacologically induced and genetic epileptic models of rodents display prominent face and construct validities [249], respectively. However, 90% of the neurological drugs with a promising preclinical potential have failed in clinical trials [250,251]. These large discrepancies could be attributed to the lack of studies on species-specific differences at the level of neural and glial cell responses to disease conditions. Recently, it was shown that apart from being small and anatomically less diverse and complex, astrocytes of rodents displayed more resilience toward oxidative stress compared to hominid astrocytes [252,253]. Unlike those of humans, mouse astrocytes display a neural repair program in response to hypoxia and no response toward glutamate agonists, whereas human astrocytes activate antigen presentation pathways in response to cytokine-induced inflammatory conditions, which contrasts with observations in mouse astrocytes. Additionally, major energy metabolic differences were found in the resting state of the mitochondrial respiration [253,254]. Therefore, more informed approaches are required for preclinical testing of antiseizure drugs in epilepsies related to neurodegeneration and refractory conditions as the underlying pathogenesis of seizures. These approaches must consider the cellular and molecular levels of altered cytokines, mitochondrial metabolism, hypoxic conditions, and functional synaptic integration, which differ in rodents and humans. Additionally, seizure generation and propagation might show considerable differences between humans and rodents due to the higher-velocity propulsion of Ca^2+^ waves in human astrocytes [252].

Although the hierarchical organization and diversity of cell types are “superficially” similar in hexalaminar cortical structures of humans and mice, ultrastructural details and high-throughput sequencing studies revealed surprising (and alarming?) differences between neurotransmitter receptors and ion channels, which constitute the major target sites of the AEDs. Notably, hominid supragranular pyramidal neurons ubiquitously expressed h-channels and had extensive dendritic arborization compared to mouse neurons, which contributed to the differences in basal electrophysiological properties and responses to high-frequency synaptic integration [255,256,257].

GWAS and transcriptomics signatures have revealed a significant correlation between human and mouse cortical microglia, but the variations were considerably enriched in host defense, cell cycle, ageing, and immune regulatory genes, as well as in genes implicated in neurodegenerative and neuro-psychiatric diseases [258,259]. These species-specific differences could contribute differentially to the tripartite crosstalk among neurons–astrocytes–microglia (Figure 3) that dominates the initial responses under *status-epilepticus* and during the process of epileptogenesis that, in turn, affect the responses of AEDs and recapitulation of human seizures during epileptic studies in rodents.

### 5.2. Informed Modeling Approaches to Tackle the Species-Specific Challenges in the Development and Testing of Epileptic Models

To tackle the challenge of species-specific biases in pro- and antiepileptic mechanisms, the inclusion of human-derived organoids, hPSCs, and humanized-mouse systems can be more informed approaches on the translational fronts of mimicking the hominid features of epilepsy disease. Each model system presents a unique set of advantages and disadvantages and can be used to address a specific set of questions.

#### 5.2.1. Human Pluripotent Stem Cells (hPSCs)

The hominid cellular- and molecular-level effects of more than 12 epileptic mutations have been modeled successfully using 2D-differentiated neural cells and glial cells from embryonic or somatic precursors [260]. Different mutations in a single gene can contribute differently to the functional consequences; therefore, patient-to-patient variability in terms of seizure types and propagation exists. The 2D model system can recapitulate the differences in mutation-specific cellular dysfunctions and guide tailor-made pharmacological or genetic treatment approaches [261]. However, the absence of extracellular neuronal and non-neuronal signaling milieu, intra-class cellular heterogeneities, and structural and functional circuitries impose limitations on the usage of this model for refractory epilepsies characterized by recurrent and spontaneous seizures as behavioral readouts [262].

#### 5.2.2. Brain Organoids

Brain organoids are 3D reconstructions of IPSCs with hominid replicas of the sub-ventricular zone, exclusive subclasses of inhibitory neurons and outer radial glial cells that are absent in rodent models [263]. Developmental epilepsies such as Angelman syndrome [264] and Rett syndrome [265] have been successfully modeled in cerebral organoids with epileptiform readouts at the level of single-cell hyperexcitability and network oscillations. High-throughput omics studies, optogenetic stimulations, and other techniques such as calcium signaling tracers and multielectrode array (MEA) chips for network-level activity have been successfully tested in organoids, rendering them as a genetically and technically amenable resource for epileptic studies [265,266,267]. Despite the high level of structural and functional relevance to the human brain, organoids are limited by the absence of vasculature, immune-extravasation, a knitted network, and a distribution of glial cells [268] and may lack vascular and immune signatures of epileptic conditions. However, studies on offering bioengineering solutions in the usage of microfluidics with endothelial-cell lining [269], the presence of astrocyte–microglial signatures in cerebral organoids [270], the incorporation of a functional vasculature-like system [271], and the structural correlation of electrophysiological properties such as burst-firing and fast-spiking of early developing neurons [272] have explored the feasibility of organoids in developmental epilepsies.

#### 5.2.3. Humanized Rodent Model

Genomic engineering and transplantation approaches can overcome the limitations imposed by the above-mentioned in vitro model systems. The grafting of disease-forming neuronal precursors (patient-derived/engineered) into the desired location of the brain of rodents allows access to the necessary neuronal inputs and interorgan communication [273,274]. A path-breaking study on the engraftment of human brain organoids was illustrated in mice by integrating axonal architecture, vascularization, neural differentiation, gliogenesis, and glial cell distribution [275]; hPSC-derived neurons of cortical origin also have been extensively transplanted into rodents’ brains [276,277]. Notably, engraftment of hPSCs derived from maturing GABAergic neurons ameliorated the epileptic activity in a pilocarpine-induced kindling model of mice [278]. These studies open avenues to the “cell therapy or (organoid-therapy?)” approaches to targeting refractory seizures and provide an amenable environment for testing of antiseizure drugs. Relevant questions regarding the “maturation of organoids in the grafted environment”, “dosage regulation”, and “critical window of grafting” remain unanswered and should be dealt with in the future.

## 6. Conclusions

Epilepsy, ID, and ASD are prevalent neurodevelopmental disorders that have proven to be complicated and challenging in several aspects. Since they often present with a highly variable and overlapping spectrum of symptoms and syndromes, defining a distinct set of diagnostic criteria has been difficult for clinicians and scientists. However, studies in mouse models of epilepsy, ID, and ASD have proved to be immensely helpful in the construction of the pathophysiology of these disorders through a bottom-top approach. These studies have demonstrated that epilepsy and ID/ASD with diverse causal origins have intersecting etiologies that might be responsible for the observed shared phenotypes. Opposing cellular phenotypes observed in these disorders highlight the importance and need for balanced and timely developmental processes at all systemic levels.

Further studies employed these aspects for the development of genetic and pharmacological therapeutic strategies (creation of mouse models with counteracting mutations to re-establish balance and the testing of drugs targeting common neurological pathways). However, only a fraction has been uncovered in the understanding of these disorders, and further studies are required for improved diagnosis, treatment, and prognosis. As highlighted in this review, there are several areas that remain unexplored and could play essential roles in the pathophysiology of epilepsy and ID/ASD. Additionally, the role of non-neuronal cells such as astrocytes, oligodendrocytes, and microglia have also not been studied in detail concerning mutations in epilepsy and ID/ASD. The augmented critical period is another characteristic modality that is altered in many forms of epilepsy and ID/ASD. A study of the precise mechanisms for a better understanding of this phase of development could be useful for the rescue during the later period of life. Because the diagnosis is delayed during the early stages of development, reversing neuronal connections becomes difficult, which is one of the significant issues lingering in the minds of neuroscientists. Preclinical studies in this regard can result in some useful clues for the translational success of the testing of small molecules for efficacy in epilepsy and ID/ASD. However, many drugs still fail in clinical trials even after ameliorating disease pathology in the preclinical mouse models. Poor experimental design with an inadequate sample size could be one of the reasons for failure at the later stages.

Another critical point is the variability in the intrinsic metabolic and biochemical pathways amongst different animal strains and species that lead to changes in drug pharmacokinetics and pharmacodynamics across systems. These factors influence how a potential therapeutic candidate molecule can be metabolized by the animal model and show how it is different for human beings. One viable alternative to overcome the above issues is to use patient-derived iPSCs, which have been considered a model in the last decade or so. However, to acquire an all-around understanding of epilepsy and ID/ASD, it is crucial to conduct studies in vivo; i.e., using animal models in combination with patient-derived iPSCs. Such combinatorial studies can fill the existing gap in our knowledge of ID/ASD and show the way toward future therapeutic strategies.

## Figures and Tables

**Figure 1 ijms-23-10807-f001:**
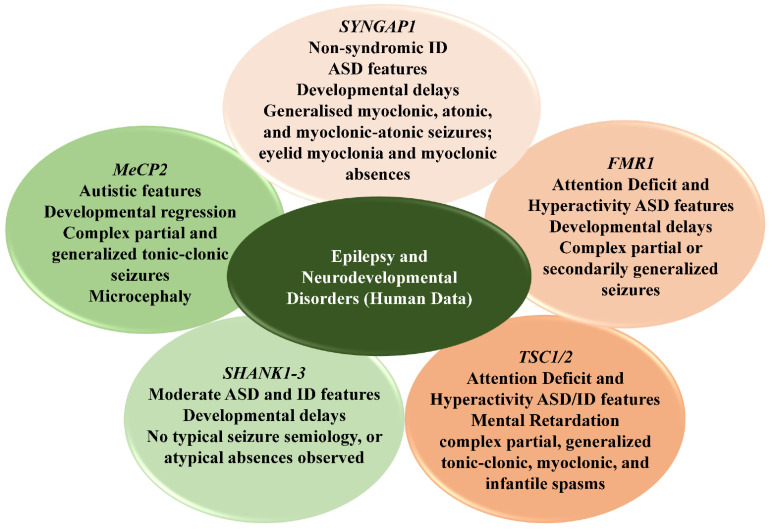
Common pathophysiological features of epilepsy in genes implicated in human NDDs. Diagram illustrating neurodevelopmental phenotypes, predominant seizure types, and electroencephalographic hallmark characteristic features observed in patient cohorts of neurodevelopmental disorders. ASD/ID, autism spectrum disorder/intellectual disability; FMR1, fragile X mental retardation protein 1; TSC1/2, tuberous sclerosis complex 1/2; SHANK1-3, SH3 and multiple ankyrin repeat domains 1-3; MeCP2, methyl CpG binding protein 2; SYNGAP1, synaptic Ras-GTPase activating protein rat homolog 1.

**Figure 2 ijms-23-10807-f002:**
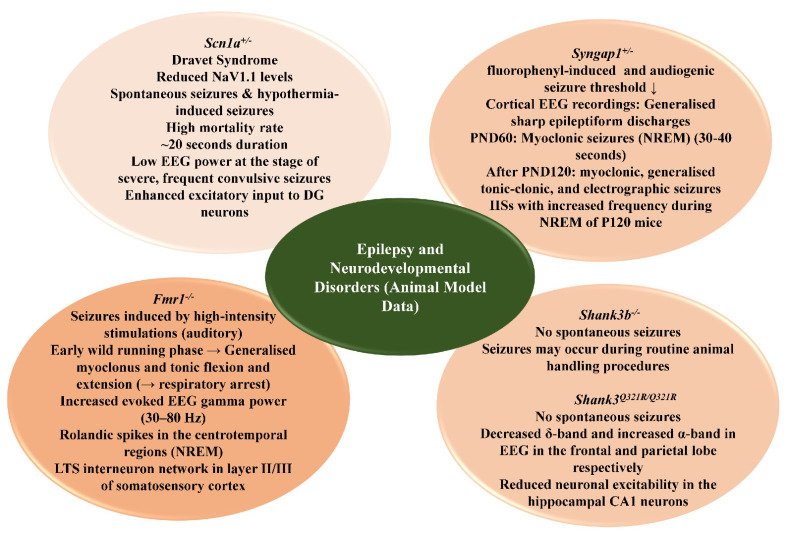
Common pathophysiological features of epilepsy in animal genetic models associated with NDDs. Diagram illustrating neurodevelopmental phenotypes, predominant seizure types, and electroencephalographic hallmark characteristic features observed in genetically developed mouse models of neurodevelopmental disorders displaying epileptic phenotypes.

**Figure 3 ijms-23-10807-f003:**
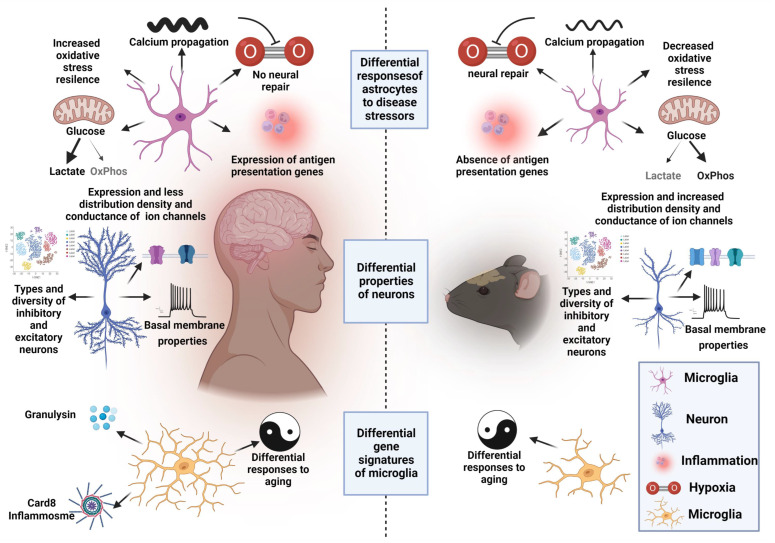
Species-specific differences in the intrinsic properties of astrocytes, neurons, and microglia. Apart from being larger in diameter and exhibiting extensive primary processes, hominid astrocytes display more resistance to oxidative stress, increased velocity of calcium propulsion, absence of enrichment in hypoxia-induced transcriptomic signatures of neural repair, inflammation-induced expression of antigen presentation genes, and resting state metabolism of mitochondria. Hominid features of morphologically complex, diverse, and larger neurons exhibit differential basal membrane properties, fewer channels, and lower ionic conductance per volume compared to rodent counterparts. Human microglia express some intrinsic gene signatures (not present in rodent microglia) such as the presence of CARD8, an inflammasome component; and granulysin, which is present in cytokine granules. Additionally, the ageing-related genes are oppositely misregulated in a species-dependent manner. Created with BioRender.com.

**Table 1 ijms-23-10807-t001:** Tabulation of epilepsy characteristics and behavioral and structural deficits in different NDD-related genes from human patient data (FMR1, SYNGAP1, SHANK, TSC, and MECP2).

Genetic Mutation	Disease	Epilepsy Characteristics	Behavioral and Structural Deficits	References
*MeCP2*	Rett syndrome (typical and atypical)	Complex partial and generalized tonic-clonic seizures. Stage I RTT has normal EEG features; Stage II RTT EEG shows loss of non-REM sleep characteristics and focal spikes or sharp waves; Stage III RTT EEG shows bilaterally synchronous bursts of pseudoperiodic delta activity and generalized rhythmic spike discharges characterizing a high seizure burden; Stage IV RTT EEG shows significant slowing of the background activity with delta rhythms, multifocal epileptiform activity in the awake state, and generalized slow spike-wave activity in sleep.	Developmental regression and delays, partial or complete loss of motor functions, gait abnormalities, abnormal sleep patterns, hand stereotypies, reduced cerebral volume and cortical grey matter mainly in frontal regions, acquired microcephaly	[59,60,61]
*SYNGAP1*	SYNGAP1-related NSID	Psychomotor delays precede epilepsy onset, seizures are mostly generalized: myoclonic, atonic, and myoclonic-atonic seizures; atypical absences; eyelid myoclonia and myoclonic absences. Ictal EEG shows generalized spike-wave discharges coinciding with the eyelid myoclonia, followed by a spike-wave complex correlating with a myoclonic (spike) and an atonic (slow-wave) component. Focal or multifocal epileptiform discharges are often observed along with generalized spike-wave discharges.	Developmental delays, language impairments, high pain and low seizure thresholds, sleeping and eating abnormalities, nonspecific MRI findings with enlarged ventricles or subarachnoid spaces, discrete hippocampal tissue loss; astrocytosis and cerebellar Purkinje neuron losses are also seen	[62]
*FMR1*	Fragile X Mental retardation syndrome	Common form of epilepsy in FXS resembles benign focal epilepsy with centrotemporal spikes (benign focal epilepsy of childhood (BFEC), benign Rolandic epilepsy). Centrotemporal spikes are the most common epileptiform feature on EEG; focal spikes or sharp discharges are seen in some patients; seizures can be partial complex and generalized tonic-clonic.	Attention-deficit/hyperactivity disorders, ASD features, aggression and self-injurious behaviors, anxiety, hand stereotypies, language deficits, regional variation in grey matter volume, linear increase in white matter volume, enlarged caudate nucleus, microcephaly	[63,64,65]
*SHANK1-3*	Phelan-McDermid syndrome	Moderate ASD/ID and refractory epilepsy of the Lennox-Gastaut type; electroencephalographic abnormalities are heterogenous: from slowing or absence of the dominant occipital rhythm to focal spike and slow-wave discharges to generalized spike and slow-wave discharges; generalised tonic-clonic, myoclonic, and tonic seizures have been reported. SHANK3 duplications can cause episodes of status epilepticus.	Developmental delay, ASD, and schizophrenia, progressive loss of skills, attention-deficit/hyperactivity disorder, dysmorphisms of corpus callosum, severe white matter alterations	[66,67]
*TSC1/2*	Tuberous sclerosis	Complex partial, generalized tonic-clonic, myoclonic, and infantile spasms characterized by multifocal EEG abnormalities	ASD/ID features, mental retardation, attention deficit, hyperactivity, aggression, anxiety, sleep disturbances, depression, altered neuronal network topology, and excitation/inhibition balance	[68]

**Table 2 ijms-23-10807-t002:** Pharmacological treatments undergoing clinical trials. This table lists the relevant clinical details of the pharmacological treatments inclusive of monotherapy and polytherapy along with their mechanism of action/actions, type of epilepsy, current clinical phase, and identifiers available on the website https://clinicaltrials.gov (accessed on: 15 July 2022).

S.No.	Intervention	Mechanism of Action	Type of Epilepsy	Clinical Phase	Clinicaltrials.gov
1	XEN1101	Potassium channel modulator	Focal onset	2	NCT03796962
2	Clobazam	Potentiation of GABAergic transmission	Refractory focal	4	NCT02726919
3	EQU-001	NA	All	2	NCT05063877
4	Cenobamate	Positive allosteric modulation of GABAA ion channels	Primary generalized tonic-clonic	3	NCT03678753
5	CX-998	T-type calcium channels	Idiopathic generalized epilepsy with absence seizures	2	NCT03406702
6	Brivaracetam	Synaptic vesicle 2A	Childhood absence	3	NCT05109234
			Rolandic benign		
				4	NCT00181116
7	YKP3089	Positive modulation of GABAA receptors and voltage-gated sodium channels	Photosensitive	2	NCT00616148
8	Ganaxolone	Allosteric to GABAA receptor	Photosensitive, drug-resistant, partial onset	2	NCT01963208
9	Prednisolone	Immunotherapy	Cryptogenic	4	NCT02695797
10	Topiramate	Inhibits carbonic anhydrase enzyme	Childhood absence	2	NCT00210574
11	BGG492	Antagonism of AMPAR	Photosensitive	2	NCT00784212
			Partial	2	NCT00887861
12	BM430C	Inhibition of voltage gated sodium channels	All	3	NCT00395694
13	Lu AG06466	Inhibits monoacylglycerol lipase (MGLL)-serine hydrolase	Focal	1	NCT05081518
14	ABI-009 (Nab-rapamycin)	Inhibition of mTOR	Surgically refractory	1	NCT03646240
15	Vorinostat	Inhibition of histone deacetylases (HDAC)	Drug-resistant	2	NCT03894826
16	ACT-709478	Inhibition of T-type Ca^2+^ channels	Photosensitive	2	NCT03239691
17	UCB0942	Antagonism of GBAA receptors	Drug-resistant focal	2	NCT02495844
18	PF-06372865	Agonism of GBAA receptors	Photosensitive	2	NCT02564029
19	VX-765	Inhibition of caspase 1	Drug-resistant partial epilepsy	2	NCT01048255
20	TAVT-18 (sirolimus)	Inhibition of mTOR	Pediatric drug-resistant	1/2	NCT04595513
21	MGCND00EP1	Modulation of 5HT1a receptors	Adolescent drug-resistant	2	NCT04406948
22	RWJ-333369	Neuromodulator	Complex partial, focal	3	NCT00433667
23	Soticlestat	Inhibition of cholesterol 24-hydroxylase	Dravet syndrome (DS) Lennox-Gastaut syndrome (LGS)	2	NCT03635073
				3	NCT04940624
24	OPC-214870	Not known	Drug-resistant	1	NCT04241965
25	TAK-935	Conversion of cholesterol to 24HC	Epileptic encephalopathies	1/2	NCT03166215
26	NBI-921352	Inhibition of Nav 1.6	SCN8A developmental and epileptic encephalopathy syndrome	2	NCT04873869
27	NBI-827104	Triple T-type calcium channel blocker	Epileptic encephalopathy	2	NCT04625101
28	LP352	5-HT2c receptor super agonist	Epileptic encephalopathy	1/2	NCT05364021
29	GWP42003-P	Cannabidiol oral solution	Dravet syndrome	3	NCT02091375
30	Ropinirole	Agonist of dopamine	Myoclonic	2	NCT00639119
31	Rufinamide	Stabilizes inactivation state of voltage-gated sodium channel	Drug-resistant	3	NCT00334958
32	Allopregnanolone injection	Positively modulates GABAA receptors	Post-traumatic	2	NCT01673828
33	Ezogabine	Positive allosteric modulation of (K(v) 7.2–7.5) channels	KCNQ2 developmental and epileptic encephalopathy	3	NCT04639310
34	Aspirin	Inhibition of mTOR	Tuberous sclerosis complex	2	NCT03356769
35	Phenylbutyrate	Removal of ammonia	STXBP1 encephalopathy	1	NCT04937062
36	STK-001	Antisense oligonucleotide to *SCN1A* mRNA	Dravet syndrome	2	NCT04740476
37	Triheptanoin	Medium-chain triglyceride	Rett syndrome	2	NCT02696044
38	Carisbamate	Moderate inhibition of high-voltage-activated calcium channels	Lennox-Gastaut syndrome	1	NCT04062981
					NCT03731715
39	CVL-865	GABAA modulation	Focal onset drug-resistant	2	NCT04244175
40	Cysteamine bitartrate (RP103)	Lysosomal metabolism of cysteine	Mitochondrial diseases, including Leigh syndrome	2	NCT02023866
41	Telampanel	Antagonism of AMPA receptors	Drug-resistant	2	NCT00057460

**Table 3 ijms-23-10807-t003:** Nonpharmacological treatments undergoing clinical trials. This table lists the relevant clinical details of the nonpharmacological treatments inclusive of gene therapies along with the potential mechanism of action/actions, type of epilepsy, current clinical phase, and identifiers available on the website https://clinicaltrials.gov (accessed on 15 July 2022).

S.No.	Intervention	Mechanism of Action	Type of Epilepsy	Clinical Phase	Clinicaltrials.gov
1	Pulvinar deep stimulation	Stimulation of pulvinar thalamic nucleus	Drug-resistant	NA	NCT04692701
2	Trans auricular vagus nerve stimulation	Experience-dependent neural plasticity	Pediatric	NA	NCT02004340
3	MRI-guided laser interstitial thermal therapy (MgLiTT)	Sinovation Laser Ablation System	Drug-resistant	NA	NCT04569071
4	Green light exposure	Engagement of thalamocortical inhibitory circuits	Drug-resistant	NA	NCT03857074
5	Fecal microbiota suspension	Modulation of gut–brain axis	Drug-resistant	3	NCT02889627
6	Bilateral thalamic central lateral nuclei stimulation	Restoration of conscious awareness	Temporal lobe	NA	NCT04897776
7	Cerebellar continuous θ burst stimulation (cTBS)	Inhibition of cortical and motor evoked potentials	Drug-resistant	NA	NCT05042726
8	Transcranial deep brain stimulation	Modulation of cortical excitability	Drug-resistant	NA	NCT04325360
9	Stereotactic laser ablation	Necrosis of epileptic foci	Temporal lobe	3	NCT02844465
10	Vagus nerve stimulation	Experience-dependent neural plasticity	Drug-resistant	1	NCT02378792
11	Lentiviral engineered potassium (K+) channel (EKC)	Gene therapy for hyperpolarization	Drug-resistant	0	NCT04601974
12	Autologous bone marrow stem cell transplantation	Tissue repair	Temporal lobe	1	NCT00916266
13	Transplantation of adipose-derived regenerative cells (ADRCs)	Tissue repair	Autoimmune drug-resistant	1	NCT03676569
14	Modified Atkins diet	Metabolism	Drug-resistant	NA	NCT01311440
15	Physical exercise program	Life-style improvement	Pediatric drug-resistant	NA	NCT05323682
16	Focused ultrasound	Modulate neuronal firing	Drug-resistant	NA	NCT03868293
17	External trigeminal nerve stimulation	Alternative to neurostimulation	Drug-resistant	2	NCT01159431
18	Vitamin D supplementation	Metabolic stimulation	Drug-resistant	3	NCT03475225
19	Betashot (a medium chain triglyceride- based (MCT) food)	Metabolic stimulation	All	NA	NCT02825745
20	Music periodicity	Musical stimulation	Benign childhood with centrotemporal spikes (BCECTS) or Rolandic	NA	NCT01515436
22	Polyunsaturated fatty acids	Anti-inflammation	Drug-resistant	3	NCT00299533

## Data Availability

Not applicable.

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
