# Peer review of "Epilepsy Characteristics in Neurodevelopmental Disorders: Research from Patient Cohorts and Animal Models Focusing on Autism Spectrum Disorder"

_ijms, 2022, doi:10.3390/ijms231810807_

Round 1
Reviewer 1 Report
This a very ambitious narrative review on a very wide topic which is epilepsy in neurodevelopmental disorders (NDDs), taking in count humans and animal models.
The title seems misleading because the Authors treat epilepsy associated with autism spectrum disorder (ASD), rather than epilepsy in comorbidity with other NDDs such as intellectual disability ID), attention deficit/hyperactivity disorder (ADHD), etc.
Furthermore, the paragraph 2.2 (Metabolic Causes) is oversimplified. There is a lot of other metabolic etiologies of epilepsy that are not mentioned here.
It is hardly understandable why the Authors treat only Environmental Causes of ASD in the paragraph 2.3. In fact, the Chapter 2 is dedicated to the Causes of Epilepsy.
Which was the main reason why the Authors choose to treat specifically a very limited number of genes, such as SCN1A, KCNA1, FMR1, TSC1/2, SYNGAP1, SHANK3, and not many other more common as etiologies of epilepsy?
The paragraphs 3.1 ( Blood and Cerebrospinal Fluid Biomarkers) and 3.2 (EEG Signatures) are poorly related with the topic of the Chapter 3, i.e. Association of Epilepsy and Neurodevelopmental Disorders (NDDs).
In synthesis, this long review appears rather unbalanced with a lot of information on epilepsy per se (which could be avoided). The Authors should rewrite this review with a different outline focusing only on genetics of epilepsy and ASD in humans and animal models.
Author Response
Response to the Reviewer-1
Manuscript Title: Epilepsy characteristics in Neurodevelopmental Disorders: Research from Patient cohorts and Animal models focussing on Autism Spectrum Disorder
We thank the reviewer for all the comments and suggestions to improve the scientific quality of the manuscript. We have addressed the comments and incorporated the suggestions raised by the reviewers in the revised manuscript. The modifications made in the revised manuscript are highlighted in yellow. The red texts indicate the reviewers' comments, and the italicised parts are the responses.
Reviewer #1: This is a very ambitious narrative review on a very wide topic which is epilepsy in neurodevelopmental disorders (NDDs), taking into count humans and animal models.
- The title seems misleading because the Authors treat epilepsy associated with autism spectrum disorder (ASD), rather than epilepsy in comorbidity with other NDDs such as intellectual disability ID), attention-deficit/hyperactivity disorder (ADHD), etc.
Response: Thank you for bringing the suggestion. We have revised the title of the manuscript to highlight the specific focus on Autism Spectrum Disorder and included the same criteria in the main text in lines 78 to line 85 of the Introduction. The revised title is as follows:
“Epilepsy characteristics in Neurodevelopmental Disorders: Research from Patient cohorts and Animal models focussing on Autism Spectrum Disorder”
- Furthermore, paragraph 2.2 (Metabolic Causes) is oversimplified. There are a lot of other metabolic etiologies of epilepsy that are not mentioned here.
Response: Thank you for bringing the suggestion. Since we have focussed on causes of epilepsy in situations where epilepsy co-presents with an ASD phenotype, we have limited our discussion of metabolic causes to only these cases. We appreciate that the metabolic basis of epilepsy is vast and warrants more elaboration, but since such an extensive description would be beyond the scope of this review, we have briefly introduced this topic and clarified a few metabolic causes of epilepsy-ASD co-occurrence. We have incorporated this suggestion and have made changes accordingly in the revised manuscript from line 154 to line 159:
There is a plethora of metabolic diseases associated with seizure occurrence, which have many mechanisms: neurotoxic ammonia accumulation in urea cycle disorders, brain energy disbalance in glucose transport disorders, and impaired ATP production in mitochondrial disorders to highlight a few. However, there are distinct patterns in situations where the epilepsy phenotype co-manifest with ASD.
- It is hardly understandable why the Authors treat only Environmental Causes of ASD in paragraph 2.3. Chapter 2 is dedicated to the Causes of Epilepsy.
Response: We thank the reviewer for the suggestion. Since we have focussed on causes of epilepsy in situations where epilepsy co-presents with an ASD phenotype, we have limited our discussion of environmental causes to only these cases. Additionally, we have reorganised and restructured the chapter on causes of epilepsy into genetic and non-genetic causes with the former further subdivided into ion-channel and receptor-mediated gene mutations and metabolic pathway mutations. We have incorporated this suggestion and have made changes accordingly in the revised manuscript from lines 170 to 175:
Environmental factors contribute to the pathophysiology of ASD and associated comorbidities that lack genetic underpinnings. Brain damage arising during delivery and through neonatal factors such as severe neonatal jaundice should be avoidable but if treated inadequately, can predispose both to epilepsy and autism. A more obvious environmental link is that of intrauterine infection. For example, maternal rubella during pregnancy has long been associated with a high risk of intellectual disability, autism and epilepsy in the offspring.
- This was the main reason the authors chose to treat a very limited number of genes, such as SCN1A, KCNA1, FMR1, TSC1/2, SYNGAP1, SHANK3, and not many other more common etiologies of epilepsy.
Response: We thank the reviewer for this question. In this review, we have chosen to elaborate on epilepsy comorbidity, specifically with neurodevelopmental disorders, particularly Autism Spectrum Disorder. The genes chosen have an established significance in being well-characterised and extensively studied animal models of Autism Spectrum Disorder. There are also several studies corroborating mouse model data with evidence of epilepsy characteristics in Autism patients with these gene mutations. This is the primary reason for choosing the genes such as FMR1, TSC1/2, SYNGAP1 etc. However, since we also chose to very briefly touch upon the conventional epilepsy models, we included some information on Scn1a mouse models which are well studied in the field of epilepsy. In addition, several studies have pointed out the autistic characteristics in Scn1a+/- mice. We have incorporated this suggestion and have made changes accordingly in the revised manuscript from lines 505 to 507:
Furthermore, several studies have investigated the autistic behaviour displayed by mice with mutated Scn1a making them relevant to the studies linking ASD and epileptic seizures.
- Paragraphs 3.1 (Blood and Cerebrospinal Fluid Biomarkers) and 3.2 (EEG Signatures) are poorly related to the topic of Chapter 3, i.e., Association of Epilepsy and Neurodevelopmental Disorders (NDDs).
Response: We thank the reviewer for suggesting this point. We agree with the reviewer and have made the necessary changes in the revised manuscript. The section on blood and cerebrospinal fluid biomarkers has been repositioned as a Box after Chapter 1: Causes of Epilepsy (added after the tables). EEG Signatures have been reorganised. We have titled chapter 2 as Patient studies of Epilepsy and ASD, and this section has been subdivided into Mechanisms of Epilepsy-ASD Comorbidity and Hallmark EEG Signatures in Epilepsy-ASD Comorbidity to maintain continuity in the text.

Reviewer 2 Report
The paper has very nice piece of information, however I have a minor comment
1. Please add a section for traumatic brain injuries, Other than chemoconvulsant models, there are animals’ model of epilepsy following Traumatic brain injury. Here are imp papers which you should cite (PMID: 30918286; PMID: 35671160; PMID: 33063874; PMID: 35065250; PMID: 34021800). Also look other papers and add a paragraph in section 5.1 (Non-Genetic model of epilepsy)
Author Response
Response to the Reviewer-2
Manuscript Title: Epilepsy characteristics in Neurodevelopmental Disorders: Research from Patient cohorts and Animal models focussing on Autism Spectrum Disorder
We thank the reviewer for all the comments and suggestions to improve the scientific quality of the manuscript. We have addressed the comments and incorporated the suggestions raised by the reviewers in the revised manuscript. The modifications made in the revised manuscript are highlighted in yellow. The red texts indicate the reviewers' comments, and the italicised parts are the responses.
Reviewer #2: The paper has a very nice piece of information, however, I have a minor comment
- Please add a section for traumatic brain injuries, Other than chemo convulsant models, there are animal models of epilepsy following Traumatic brain injury. Here are imp papers which you should cite (PMID: 30918286; PMID: 35671160; PMID: 33063874; PMID: 35065250; PMID: 34021800). Also, look at other papers and add a paragraph in section 5.1 (Non-Genetic model of epilepsy)
Response: We thank the reviewer for suggesting this point. We agree with the reviewer and have made the necessary changes in the revised manuscript from lines 435 to 458 as given below:
Traumatic Brain Injury
Mild to moderate brain injuries can lead to complications of seizures, and in some cases, the spontaneous reoccurrence of seizures develops into post-traumatic epilepsy (PTE). Animal models of PTE are essential for understanding the pathophysiology of the resultant epileptic seizures as the condition is prone to pharmacoresistance, increasing the necessity for effective therapeutic strategies. Some of the existing animal models of PTE include the Fluid Percussion Injury model, Controlled Cortical Impact (CCI) model, Impact Acceleration Model, Canine Model of Post-Traumatic Epilepsy, Penetrating Head Trauma Model and Paediatric Post-Traumatic Epilepsy model which replicates the neuroinflammatory, metabolic and neurodegenerative characteristics of PTE patients. Among these, the most extensively studied model is the Fluid Percussion Injury model, which is developed by applying pressure pulses of 0.9–2.1 atm (1.5-3.4 atm for rats) for approximately 20 ms after performing craniotomy over the right parietal cortex keeping the dura intact in mice. CCI models of rats and mice are also important models for PTE created by causing cortical malformations in the exposed brain by utilising pneumatic or electromagnetic impactors at different velocities for varying severity. Ablation of brain injury centred over the left or the right sensorimotor cortex created asymmetrical responses. Another model developed involved the administration of homocysteine (845 mg/kg, intraperitoneal) 16–18 hours after cobalt implantation, resulting in refractory cortical-onset SE accompanied by injury, with evidence of widespread neocortical oedema and damage. This is similar to the conditions observed in SE arising from TBI, subarachnoid haemorrhage, and lobar haemorrhage. A two-hit model for epilepsy has also been developed by utilising optogenetic kindling where a secondary factor of injury and inflammation can be added for inducing the spontaneous seizure allowing the investigation of injury or inflammation's role in epileptogenesis without being hindered by electrode insertion as in case of electrical kindling.

Round 2
Reviewer 1 Report
The Authors modified all the unclear points in this paper, and it is rather improved in the new version.